# Abdominal Cryptorchidism with Complete Dissociation between the Testis and Deferent Duct Mimicking Testicular Regression Syndrome

**DOI:** 10.3390/children10020205

**Published:** 2023-01-23

**Authors:** Vladimir V. Sizonov, Alexey G. Makarov, Johannes M. Mayr, Vladimir V. Vigera, Mikhail I. Kogan

**Affiliations:** 1Division of Pediatric Urology and Andrology of the Department of Uroandrology and Human Reproductive Health, 344022 Rostov-on-Don, Russia; 2Department of Uroandrology, Regional Children’s Clinical Hospital, 344015 Ros-tov-on-Don, Russia; 3Department of Pediatric Surgery, University Children’s Hospital Basel, Spitalstrasse 33, 4056 Basel, Switzerland

**Keywords:** abdominal cryptorchidism, dissociation, epididymis, deferent duct, malformation, testicular regression syndrome

## Abstract

Complete separation of the deferent duct from the epididymis in cryptorchid testes residing in the abdomen is an extremely rare variant of developmental disorders of the testis and epididymis. Available sources mention only three clinical cases similar to our observations. The unique anatomic aspects of this disorder hamper the correct diagnosis of an intra-abdominal cryptorchid testis. Two boys with nonpalpable left-sided cryptorchidism underwent diagnostic laparoscopy, revealing an intra-abdominally located testis. The epididymis was completely separated from the deferent duct, and the epididymis and testis were supplied by testicular vessels. Exploration of the inguinal canal revealed blind-ending deferent ducts. The testis was brought down through the inguinal canal and fixed in the scrotum in both boys. The follow-up examination at 6 months revealed no signs of testicular atrophy or malposition of the testis in either patient. With our observations in mind, the exclusive use of a transscrotal or transinguinal approach as the initial surgical exploration in the treatment of patients with nonpalpable forms of cryptorchidism may be inappropriate. Careful laparoscopic examination of the abdominal cavity is indispensable in children with suspected testicular regression syndrome or nonpalpable forms of cryptorchidism.

## 1. Introduction

The incidence of cryptorchidism among full-term newborns ranges from 1.0 to 4.6% at the time of birth [1]. Palpable forms of cryptorchidism are present in 73 to 90% of affected boys, while the testicle resides in a nonpalpable position in 10 to 27% of patients [2,3,4,5,6,7]. In half of the patients suffering from nonpalpable cryptorchid testes, the testes are located intra-abdominally. Most of the remaining patients suffer from testicular regression syndrome. Surgical intervention in cryptorchidism aims to correct the position of the cryptorchid testis, and the deferent duct can serve as a guiding structure when locating the testicle during surgery [8,9,10,11,12].

Virilization of the male genital system starts around the 7th week of gestation. The spermatogenic cord develops in the fetal testis, and the Mullerian duct disappears. At 12 weeks of gestation, cords from the rete testis give rise to the efferent ducts, which fuse with the mesonephric tubules close to the testis to form the epididymal duct. The vas deferens develops from the proximal vas precursor, which represents the central portion of the mesonephric duct [13].

Congenital absence of the vas deferens (CAVD) occurs in 0.1% of men. In the majority of men suffering from CAVD, at least one cystic fibrosis-causing gene mutation is found [14]. However, in 10 to 20% of cases of congenital bilateral absence of the vas deferens and in 60 to 70% of men with congenital unilateral absence of the vas deferens, no genetic mutations could be found [14]. In many of these unexplained CAVD cases, a solitary kidney is found. Unilateral renal agenesis occurs in 5 to 10% of patients suffering from congenital bilateral absence of the vas deferens and in 20 to 40% of patients with congenital unilateral absence of the vas deferens [14].

Undescended testes are frequently associated with fusional anomalies of the testis and epididymis, and these malformations are nowadays made responsible for infertility in men suffering from undescended testes [15].

Complete separation of the deferent duct from the epididymis is an extremely rare variation of developmental disorders of the testis, epididymis, and deferent duct. Failure of fusion of the epididymis and testis with complete separation of the epididymis has rarely been described in the literature [16,17,18,19]. In both of our patients, the failure of fusion occurred at the level of the junction between the tail of the epididymis and the vas deferens.

These unique anatomic aspects may make it difficult to locate the intra-abdominal testis. Thus, we feel that the usual surgical approach based on the assumption that the connection between the epididymis and its deferent duct is inseparable requires some correction.

We aimed to make pediatric surgeons and pediatric urologists aware of this very rare disorder characterized by separation of the epididymitis and deferent duct in cryptorchid testes located abdominally. Unidentified testes may remain in the abdominal cavity rather than being transferred to the scrotum.

## 2. Case Description

Patient 1, aged 21 months, suffered from nonpalpable cryptorchidism. During diagnostic laparoscopy, we noted that the internal ring of the left inguinal canal was obliterated and the deferent duct with hypoplastic “testicular” vessels entered the internal inguinal ring (Figure 1).

We first established the diagnosis of testicular regression syndrome and planned to perform transinguinal exploration to search for the so-called “testicular lump”. During further diagnostic laparoscopy, we noted an ipsilaterally located abdominal testis and epididymis, which were located in the small pelvis on a vascular pedicle but were detached from the deferent duct (Figure 2).

The distance between the internal inguinal ring and the crossing of testicular vessels from the retroperitoneal to the intraperitoneal positions was approximately 4 cm. 

We opted for orchiopexy because the testicular vessels appeared to be sufficiently long. An incision was made in the left inguinal area to open the inguinal canal. Exploration of the inguinal canal revealed a blind-ending deferent duct (Figure 3A,B).

The left testis was brought down through the external inguinal ring into the upper third of the scrotum and fixed using the Shoemaker’s technique (Figure 4).

Patient 2, aged 48 months, underwent diagnostic laparoscopy for a nonpalpable left-sided testis. Laparoscopy revealed an intra-abdominal testis and epididymis attached to the testicular vessels. The testis and epididymis were completely separated from the deferent duct. The distance between the internal inguinal ring and the crossing of testicular vessels from the retroperitoneal space to the intraperitoneal position was 4 mm to 5 mm, and the processus vaginalis was patent (Figure 5).

Exploration of the inguinal canal revealed a blind-ending deferent duct (arrow; Figure 6).

The testis was brought down through the external inguinal ring into the middle third of the scrotum and fixed using Shoemaker’s technique. No intra-operative complications were encountered, and the postoperative course was uneventful. Both patients were discharged on day 2 after surgery, and the follow-up examination at 6 months revealed no signs of testicular atrophy or malposition in either patient.

None of our patients showed any signs of cystic fibrosis or congenital renal agenesis.

## 3. Discussion

Anomalies of connection between the testis, epididymis, and deferent duct are associated with cryptorchidism in most cases [20]. In the 1990s, a series of studies categorized various types of testicular and epididymal developmental disorders [13,21,22,23]. Vohra et al. provided a detailed description and classified anomalies of the deferent duct, epididymis, and seminal vesicles [13]. However, none of the classifications described a complete separation of the deferent duct and epididymis.

The embryogenesis of the urogenital system must be well understood to appreciate the mechanism of development of this anomaly. The testis and deferent duct originate from different embryologic structures. The head of the epididymis and testis develop from the urogenital ridge, consisting of the genital and mesonephric ridges. The efferent ducts and rete testis develop from the mesonephric tubules. The epididymal duct and the deferent duct develop from the Wolffian (mesonephric) duct [17,19]. Complete or partial dissociation of these two systems leads to various anomalies, in particular the complete separation of the deferent duct and epididymis observed in our two patients.

To locate the testis in nonpalpable forms of cryptorchidism, two surgical approaches are used. In the absence of vicarious hypertrophy of the contralateral testis, diagnostic laparoscopy is the method of choice. During laparoscopy, surgeons may mistakenly interpret the deferent duct entering the “closed” deep inguinal ring as testicular regression syndrome. As a result, surgeons may opt for the transinguinal approach to search for the testis. The blind end of the vas deferens detected in the inguinal canal may be erroneously interpreted as a testicular remnant. In turn, misleading intraoperative findings may result in the testicle being left in the abdomen.

In accordance with the findings of Barthold and Redman, the partial agenesis of the vas deferens was associated with cryptorchidism in our cases [24]. Male duct agenesis in a child is an extremely rare event. In most cases, the testis is of normal size and exhibits unimpaired hormonal function. Therefore, the testis should be preserved [25]. We undertook one-stage orchiopexy in both patients and successfully placed the testes in the scrotum.

Abdelmohsen et al. described a patient whose distal part of the right vas deferens was missing [26]. The authors pointed out that no one had attempted to anastomose the vas deferens to the epididymis in similar cases so far [26].

Bilateral congenital absence of the vas deferens was considered a virtually untreatable cause of male infertility before microsurgical aspiration of sperm from the epididymis and vasa efferentia for in vitro fertilization and embryo transfer, as described by Silber et al. [27], became possible. Mickle et al. investigated 21 infertile men suffering from congenital unilateral absence of the vas deferens and studied mutations in the cystic fibrosis genes [28]. Among the 12 men who had patent and anatomically complete contralateral vasa deferens, none exhibited a mutation of the cystic fibrosis gene. In contrast, 8 of 9 patients suffering from noniatrogenic occlusion of the contralateral vasa deferens carried a mutation of the cystic fibrosis gene [28].

Although our clinical observations represent a very rare anomaly in the development of the testis, epididymis, and vas deferens based on the limited literature evidence, the stereotypes of laparoscopic images should be challenged to avoid diagnostic errors. Thus, a thorough laparoscopic search for the nonpalpable gonad is indispensable.

An alternative surgical technique to locate a nonpalpable gonad involves the transscrotal or transinguinal approach as the starting option. Since 1969, several authors have published articles on the role of vicarious testicular hypertrophy in unilateral, nonpalpable cryptorchidism as an indicator of the state of the cryptorchid gonad. This procedure assumed that hypotrophy or atrophy of an organ that exists as a pair, such as the ovaries, kidneys, and adrenal glands, would lead to compensatory hypertrophy of the contralateral healthy organ. Studies revealed that the hypertrophy of a healthy testis completely compensates for the absence, hypotrophy, or atrophy of the contralateral testis. Hurwitz et al. found that in 90.3% of patients with unilateral nonpalpable testis, an increase in the testicular size of the contralateral testis to 1.8 cm or more was indicative of the missing testis [29]. Snodgrass et al. reported that a testicular size exceeding 1.8 cm to 2.0 cm constitutes a predictive sign of monorchism in 88% of patients [30]. Shibata et al. reported that the absence of a testicle is highly likely if the length of the healthy testis exceeds 22.4 mm and the testicular volume exceeds 2.2 mL [31]. Similarly, Braga et al. concluded that monorchism is associated with a testis length of more than 19 mm on the contralateral side [32]. According to Hodhod et al., testicular hypotrophy or atrophy is detected with 100% reliability in the presence of compensatory hypertrophy of the contralateral healthy testis to greater than 2.0 mL [33]. Consequently, an approach has been developed in which vicarious hypertrophy of the contralateral testis is considered an indication for surgical exploration of the scrotal and inguinal regions.

When using the transscrotal or transinguinal approach in children with vicarious hypertrophy of the contralateral gonad, surgeons should be aware of the rare developmental anomaly described in this article. Inspection of the inguinal canal may reveal a vas deferens with a small lump at the end (hypotrophic body and tail of the epididymis), which may be misinterpreted as a testicular remnant.

Thus, the question arises whether we may have treated certain patients for suspected testicular regression syndrome in the past and overlooked the intra-abdominally located gonad in certain boys who suffered from dissociation of the deferent duct from the epididymis associated with cryptorchidism. This diagnostic error may have been the consequence of our former conviction that the testis and epididymis cannot exist without being connected to the deferent duct. 

## 4. Conclusions

Although complete dissociation of the epididymis and vas deferens as described here is a very rare anomaly in the development of the epididymis and vas deferens, the small likelihood of its occurrence should be borne in mind when treating patients with nonpalpable forms of cryptorchidism.

With our observations in mind, the exclusive use of a transscrotal or transinguinal approach as the initial surgical exploration in the treatment of patients with nonpalpable forms of cryptorchidism may be inappropriate. Careful laparoscopic examination of the abdominal cavity is indispensable in children with suspected testicular regression syndrome or nonpalpable forms of cryptorchidism.

## Figures and Tables

**Figure 1 children-10-00205-f001:**
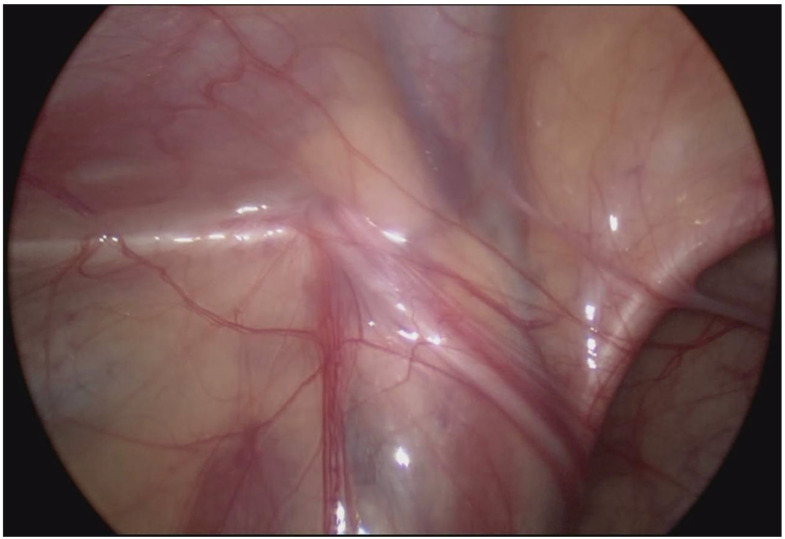
Deferent duct with hypoplastic “testicular” vessels entering the inguinal canal.

**Figure 2 children-10-00205-f002:**
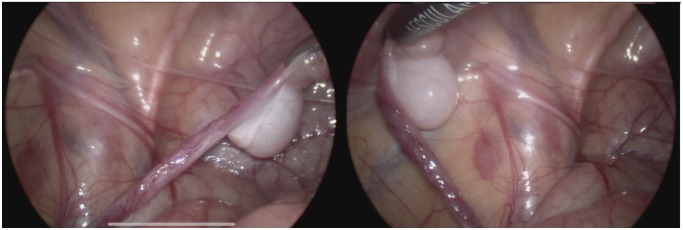
Testis with epididymis attached to testicular vessels but disconnected from the deferent duct.

**Figure 3 children-10-00205-f003:**
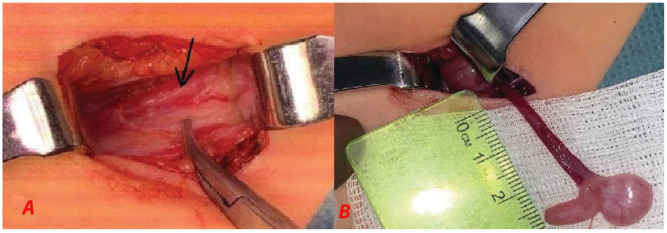
(**A**) The inguinal canal was opened. The arrow indicates the blind-ending deferent duct. (**B**) Testis with epididymis attached to testicular vessels.

**Figure 4 children-10-00205-f004:**
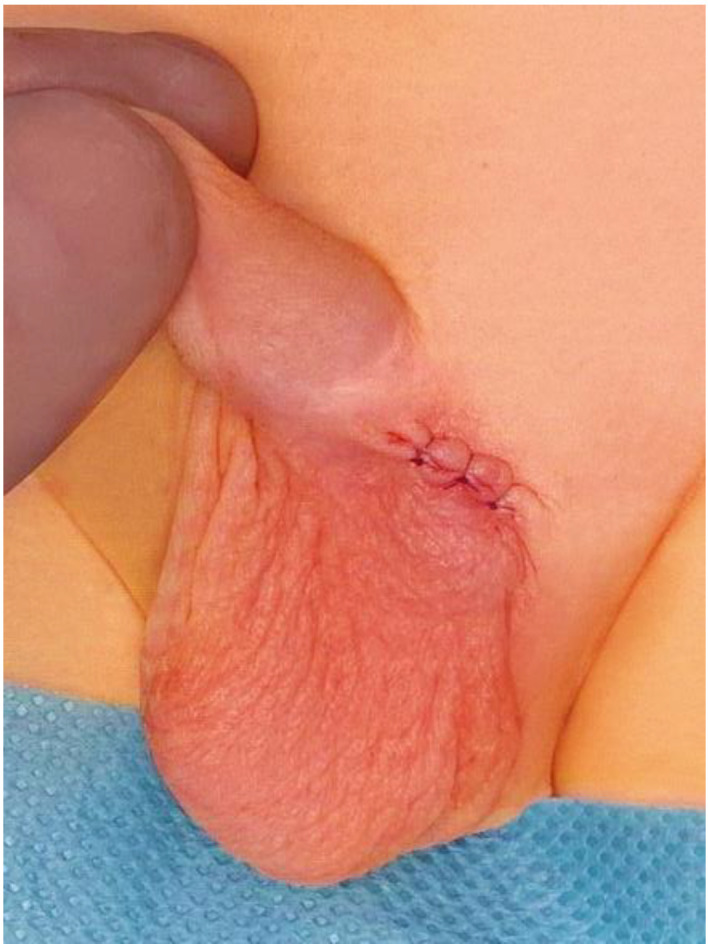
The left testis was fixed in the upper scrotum.

**Figure 5 children-10-00205-f005:**
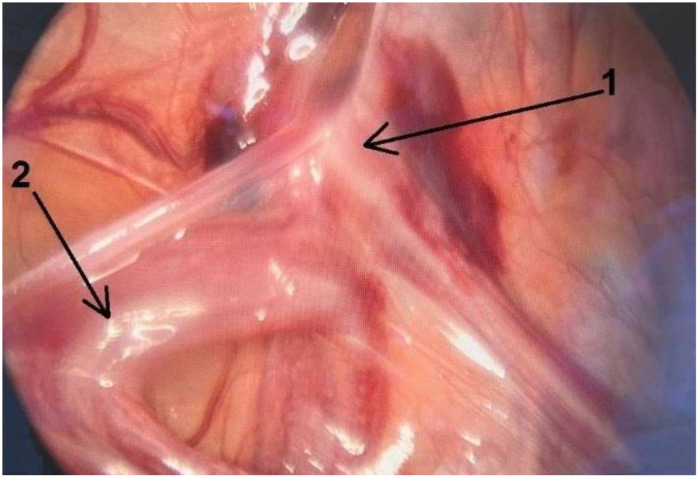
Deferent duct (**1**) entered the inguinal canal. The testis with epididymis (**2**) was connected to testicular vessels but was disconnected from the deferent duct.

**Figure 6 children-10-00205-f006:**
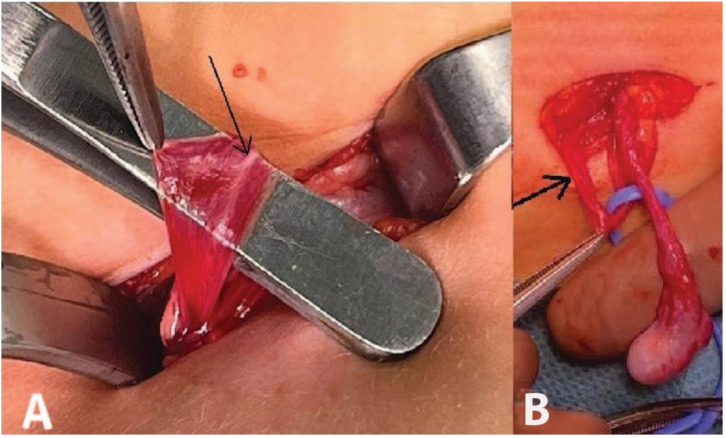
(**A**) The inguinal canal was opened. The deferent duct appeared blind-ended. (**B**) The arrow indicates the deferent duct. The tip of the forceps indicates the blind end of the deferent duct. The epididymis was not connected to the deferent duct.

## Data Availability

No new data were generated for this retrospective case report.

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
