# Peer review of "Abdominal Cryptorchidism with Complete Dissociation between the Testis and Deferent Duct Mimicking Testicular Regression Syndrome"

_children, 2023, doi:10.3390/children10020205_

Round 1
Reviewer 1 Report
Epididymis anatomy which is abnormal muste be described, double vas defferens unilateraly is nor extremely rare condition. Therefore literature search should be extended. For exmple Gravgaard et al 1978
Author Response
Answer to Reviewer 1:
We thank Rev. 1 for the careful review of our manuscript and we extended the literature search (new references are marked in red) and discussed these references in the text (page 14).
We also described the most important congenital anomalies of the epididymis and vas deferens (page 15).
We also included the article of Gravgaard et al. (page 14, ref. 17)
Reviewer 2 Report
1. This is a rare presentation of testicular and epididymal fusion anamoly in the from of complete separation of vas deferens from epididymis , reference quoted (13-15) for similar anomalies are not similar.
El Gohary reported a complete testicular epididymal dissociation.
other two has findings where proximal epididymis is attached to the testis and another part with vas deferens .
2. Discussion need to be based on rare finding of complete separation of vas deferens with epididymis.
3. conclusion need to be more precise in pertaining to rare findings.
4. For embryological development of testis epididymis and vas- appropriate references must be quoted
Author Response
- This is a rare presentation of testicular and epididymal fusion anamoly in the from of complete separation of vas deferens from epididymis , reference quoted (13-15) for similar anomalies are not similar.
We thank Rev. 2 for their thoughtful comments which helped us to revise the manuscript. We agree with reviewer 2 and revised the text accordingly (page 3; ref.: 19-22),
The English and style of the article was revised by Silvia M. Rogers, PhD, of MediWrite, Basel, Switzerland.
El Gohary reported a complete testicular epididymal dissociation.
We agree with reviewer 2 and revised the text as suggested by reviewer 2 (page 3 ref. 22)
other two has findings where proximal epididymis is attached to the testis and another part with vas deferens .
This has been corrected (page 3).
- Discussion need to be based on rare finding of complete separation of vas deferens with epididymis.
We included a discussion of the complete separation of vas deferens from epididymis in the discussion section (page 7-8)
- conclusion need to be more precise in pertaining to rare findings.
We revised the conclusion (page 10)
- For embryological development of testis epididymis and vas- appropriate references must be quoted
We cited more references (page14-15; ref.: 16,18,25,26) and added a brief description of the development of the epididymis and vas (page 2 ref. 15,16).
Reviewer 3 Report
Thank you for having opportunity to review this manuscript.
I have to confess that I liked the manuscript.
Dealing with undescended testis in childhood is still debated in many aspests.
One of these debated issues is the treatment of vanishing testis ( testicular regression syndrome , TRS).
The cases, presented by the authors give interesting further knowledge to this topic. Although very rare variants, the presented cases drew attention to the differential diagnosis of nonpalpable forms of cryptorchidism.
The introduction is appropriate, the text is clear and easy to read. The figures, pictures are informative and of good quality. References are significant and relevant.
I have two minor remarks, suggestions:
1.In the abstract and later in the text , authors conclude that " We hypothesize ,that our observations significantly limit the use of transinguinal revision as first line surgery when aiming to locate the testes in nonpalpable forms of cryptorchidism." I would rather use transinguinal approach, insted of revision. Revision is used more widely for secondary surgeries or re-operations.
2. The main conclusion - in my opinion- is , that "careful laparoscopic examination of the abdominal cavity .....is indispensable. " I would emphasize this conclusion more markedly.
Author Response
We thank Rev. 3 for their encouraging comments.
I have two minor remarks, suggestions:
1.In the abstract and later in the text , authors conclude that " We hypothesize ,that our observations significantly limit the use of transinguinal revision as first line surgery when aiming to locate the testes in nonpalpable forms of cryptorchidism." I would rather use transinguinal approach, insted of revision. Revision is used more widely for secondary surgeries or re-operations.
We thank Rev. 3 for their suggestion and replaced the word “revision” accordingly.(page 1 and entire manuscript)
- The main conclusion - in my opinion- is , that "careful laparoscopic examination of the abdominal cavity .....is indispensable. " I would emphasize this conclusion more markedly.
We agree with Rev. 3 and changed the conclusion accordingly. (page 1 and 10)
Round 2
Reviewer 2 Report
References related to Poly vas deferentia or duplication are unrelated to the text.
Author Response
Comments of Rev. 2
References related to poly vas deferentia or duplication are unrelated to the text and should be deleted.
Response to Reviewer 2
We thank Rev. 2 for their careful review and thoughtful comment and deleted the text and references related to the topic of poly vas deferens and duplication of the vas deferens. We marked changes to the manuscript in red.